# Identification of Micro-dynamics Phase Transition Processes for Ammonium Sulfate Aerosols by Two-dimensional Correlation Spectroscopy

Xiuli Wei[1,2], Xiaofeng Lu[1,2], Huaqiao Gui [1,2,3*], Jie Wang[1], Dexia Wu[1], Jianguo Liu[1,2]

1 Key Laboratory of Environmental Optics and Technology, Anhui Institute of Optics and Fine Mechanics, Hefei Institutes of Physical Science, Chinese Academy of Sciences, Hefei 230031, China

2 School of Environmental Science and Optoeclectronic Technology, University of Science and Technology of China, Hefei, 230026, China

3 Institute of Environmental Hefei Comprehensive National Science Center, Hefei 230088, China

* Correspondence: Huaqiao Gui (hqgui@aiofm.ac.cn)

## Abstract

Phase transitions of particles are importance because it could influence reactive gas uptake, multiphase chemical reactions pathways, ice and polar stratospheric cloud formation. Electron microscopy-based techniques detect phase transitions through macroscopic properties but cannot resolve efflorescence kinetics at the molecular level, whereas spectroscopic methods, while enabling molecular-level tracking of aerosol transformations and water uptake quantification, remain limited in characterizing the dynamics of efflorescence phase transition due to insufficient temporal resolution to resolve intermolecular interactions and capture complex intermediate states. In this study, the efflorescence phase transition of AS aerosol is tracked by using the fourier transform infrared (FTIR) spectroscopic technique. A novel method has been developed that not only accurately determines the phase transition region but also enables the identification of micro-dynamic phase transition processes through analysis of the molecular structural change sequence based on two-dimensional correlation spectroscopy. During efflorescence transition processes, we measured the phase transition point at $39\% \pm 0.8\%$ (RH), and its start and end points at $41\% \pm 0.8\%$ (RH) and $36\% \pm 0.8\%$ (RH), respectively. We further identified four distinct micro-dynamics steps during the efflorescence processes. Initially, there was a gradual loss of liquid water for the solution droplets. Subsequently, it formed the supersaturated ammonium sulfate (AS) aerosols. Furthermore, hydrogen bonds between liquid water and sulfate dissociate, reducing aqueous sulfate concentration. The replacement of $H_2O$ by $NH_4^+$ in the sulfate hydration shell drives a bond reorganization that strengthens sulfate-ammonium interactions at the expense of original sulfate-water bonds. The

efflorescence occurs and forms crystal/solid AS. Eventually, the remaining liquid water molecules eventually detach from the AS system, completing the liquid-to-solid phase transition. This method will help improve comprehending of the transport and deposition of inhaled aerosol. Moreover, these insights will spur fundamental research into the formation, transformation and nucleation mechanisms of atmospheric aerosols.

**Keywords:**

phase transitions

micro-dynamics mechanism

efflorescence processes

supersaturated ammonium sulfate

## **Introduction**

The phase state of aerosols governs their physical, chemical, and optical properties, thereby exerting significant impacts on the environment, climate system, and human health(Shiraiwa et al., 2017) (Meng et al., 2024), (Poschl, 2005) . It alters the phase of temperature changes in addition to the effects of relative humidity and particle size effects.(Martin, 2000; Shiraiwa et al., 2017),(Xie et al., 2017). Phase transitions of aerosol could affect reactive gas uptake, multiphase chemical reactions pathway, ice and polar stratospheric cloud formation (Martin, 2000).

It has long been assumed that aerosol phase transition behaviors are thermodynamic equilibrium and may be achieved in a short amount of time. For example, the deliquescence of particulate matter is one of phase transition behaviors. The particle should take up some amount of water to establish thermodynamic equilibrium upon increasing humidity. Currently, the phase transition behaviors of aerosol have been extensively investigated by the environmental scanning electron microscope (ESEM)(Treuel, Pederzani, & Zellner, 2009; Wise, Martin, Russell, & Buseck, 2008), the hygroscopic tandem differential mobility analyzer (H-TDMA)(Gao, Chen, & Yu, 2006), the electrodynamic balance (EDB)(Cohen, Flagan, & Seinfeld, 1987), microresonator mass sensor(Zielinski et al., 2018), and optical microscopy(Ciobanu, Marcolli, Krieger, Zuend, & Peter, 2010) and other methods. These methods can diagnose the phase transition processes of particles by measuring the particle size, shape and other physical parameters. They have been recognized as some main parameters during the phase transition processes of aerosol particles(Cheng,

Su, Koop, Mikhailov, & Poschl, 2015; Gao et al., 2006). These findings reveal that water uptake, initiated by phase transitions in complex aerosols (Wise et al., 2008), leads to organic-inorganic phase separation in fine particles (W. Li et al., 2021), thereby providing reactive surfaces that facilitate secondary aerosol formation(Sun et al., 2018).

However, it is increasingly recognized that kinetic mechanisms - including molecular diffusion, ion pairing, hydration-shell reorganization, and nucleation barriers - play a crucial role in driving deviations from equilibrium predictions, leading to phenomena like delayed efflorescence and humidity hysteresis that cannot be fully resolved by conventional physical measurements.

Efflorescence, as a kinetically controlled process, involves gradual involves the gradual evaporation of salt-laden water and a solid-state transition that begins with crystal nucleation under supersaturation (Karthika, Radhakrishnan, & Kalaichelvi, 2016; Miles et al., 2025). On molecular level, such phase transition processes are not thermodynamic equilibrium, since molecular-scale dynamics inherently deviate from thermodynamic equilibrium conditions. The interaction with water vapor induces modifications in both the composition and local chemical microenvironment of aerosols. Conventional Fourier transform infrared (FTIR) spectroscopy can be utilized to observe these transformations and quantify the water uptake of aerosols, though its capability to resolve aerosol evaporation kinetics remains limited (S.-S. Ma, Yang, Zheng, Pang, & Zhang, 2019; S. Ma et al., 2024). Moreover, the overlapped and time resolution restrict its development to be a certain range in the hygroscopic or nucleation property of aerosols. In order to study the complicated physical or chemical transition processes of aerosol, two-dimensional correlation infrared spectroscopy (2D-IR) has been used (Wei et al., 2022), (Chen, Teng, Qian, & Yu, 2019). Nevertheless, the molecular-scale dynamics controlling aerosol efflorescence remain inadequately characterized, largely due to the difficulty of directly probing transient molecular intermediates and spatially heterogeneous transitions under atmospherically relevant conditions.

To address these challenges, a precise analytical methodology was developed to track complex spectral changes and investigate the micro-dynamics phase transition mechanism of aerosols. During the efflorescence transition processes, we examined the intermolecular interactions and identified phase change points by coupling a controlled relative humidity (RH) system and 2D-IR. This approach provides further insights into

101 the formation and transformation mechanisms of atmospheric aerosols.

# 1. Instruments and Methods

## 1.1 The samples and measurement System

**Samples:** In this study, all aerosol samples were generated from Ammonium Sulfate (AS) solution using an aerosol generator. The concentration of AS solution was 4.0 g/L.

**The experiment system**:The experiment system mainly includes a humidification system and an in situ FTIR system which has been described in ACP (Wei et al., 2022). The experimental setup, including the atomizer, dryer, and DMA layout, was identical to that described by Wei et al (Wei et al., 2022). The phase transition processes of the aerosol were measured by transmission FTIR spectroscopy (Tensor 27, Bruker Optics, Germany). One end of the sample cell is provided with a radius of 3 cm zinc selenide (ZnSe) substrate and the other end is the same ZnSe substrate containing aerosol samples. The spectral resolution is $4 \, cm^{-1}$ and a repeat time of 1 scan. The humidification system is used to provide a certain RH for the aerosol samples. It consists of dry and humidified $N_2$. The humidified $N_2$ was supplied by the high purity water vapour. By adjusting the volumetric ratio between these two $N_2$ streams, we could obtain a specific RH. Its accuracy is $\pm 0.8 \%$ for a $0\sim100 \%$ RH range, and its time resolution is about 30 second. The FTIR spectrometer starts to take absorption spectra of the samples approximately 1 min after the injection of each designated RH. This time interval is used to stabilize the RH inside the sample cell. We measured the RH at the outlet of the cell and confirmed that the difference from the inlet RH did not exceed 0.5% (RH). The AS aerosols were humidified or dehumidified at a rate of 1% $min^{-1}$ within the range of 20% to 90% (RH).

In this study, the aerosol samples with electrical mobility diameter about 300 nm were sorted into a specific electrical mobility diameter ($D_{em}$) by a differential mobility analyzer (DMA; model 3082, TSI). And they were deposited on ZnSe substrate. The mass of the deposited AS is 0.074 mg (Fig.S1) which was measured by Ion chromatograms (ICS-3000, Dionex, United States) and the area loading on ZnSe is about 0.79 $cm^2$. Since this area is smaller than the infrared spot illuminating the sample, the acquired infrared spectrum represents the total absorption spectrum of AS. The deposited particulate film had a thickness equivalent to approximately three particle monolayers. The 300nm particles may overlap and form stacks on the substrate during

deposition. However, FTIR analysis verifies that this physical arrangement does not
change their chemical composition.

## 1.2 The two-dimensional correlation infrared spectroscopy analysis method

### 1.2.1 Generalized two-dimensional infrared (2D-IR) correlation spectroscopy

The two-dimensional (2D) correlation spectral function can be represented as follows:

$$X(\nu_1, \nu_2) = <\tilde{y}(\nu_1, t) \cdot \tilde{y}(\nu_2, t)> = \Phi(\nu_1, \nu_2) + i\Psi(\nu_1, \nu_2)$$

Here, $\tilde{y}$ represents a set of the dynamic spectra that are functions of both spectral
variables ($\nu_1$ and $\nu_2$, corresponding to the spectral wavenumber of compounds the
vibrations wavenumbers) and the external perturbation variable (relative humidity,
RH). The synchronous ($\Phi(\nu_1, \nu_2)$) and asynchronous ($\Psi(\nu_1, \nu_2)$) correlation
intensities, corresponding respectively to the real and imaginary components of the
complex cross-correlation function, quantitatively describe the coordinated and
sequential changes in spectral intensities at wavenumbers $\nu_1$ and $\nu_2$ (Chen et al., 2019;
Isao Noda & Ozaki, 2014). In this study, we use the synchronous correlation maps to
diagnose if the spectral intensities at different wavenumbers vary simultaneously, and
the asynchronous correlation maps are used to identify the occurrence sequential order
of the intermolecular interactions. In the synchronous correlation maps, the positive
and negative correlations indicate simultaneous and opposite changes of the spectral
intensities observed at the wavenumber pair ($\nu_1$, $\nu_2$), respectively. In asynchronous
correlation map, positive cross-peaks suggest that spectral intensity variations at
frequency $\nu_1$ precede those at $\nu_2$, whereas negative cross-peaks imply the inverse
temporal sequence, with changes at $\nu_2$ leading those at $\nu_1$.

### 1.2.2 Perturbation-correlation moving window two-dimensional (PCMW2D) correlation infrared spectroscopy

PCMW2D correlation infrared spectroscopy serves as a powerful tool for recognizing
characteristic spectral variation along the perturbation variable axis(Isao Noda, 2025;
I. Noda, Park, & Jung, 2025), (Morita, Shinzawa H Fau - Noda, Noda I Fau - Ozaki, &
Ozaki, 2006). This technique generates complementary synchronous and asynchronous
2D correlation spectra, visualized as contour maps with spectral variables (e.g.,
wavenumber) plotted against perturbation parameters (e.g., relative humidity),
enabling precise identification of transition points and critical regions. Consequently,
PCMW2D proves particularly valuable for elucidating complex environmental

processes through its distinctive analytical capabilities. In the resulting 2D-IR spectra, red-colored regions indicate positive correlation intensities, whereas blue-colored regions denote negative correlation intensities. Within synchronous correlation maps, these positive and negative correlations correspond to enhanced and diminished spectral intensity variations along the perturbation gradient, respectively. The asynchronous spectra reveal more nuanced behavior: positive correlations signify convex spectral intensity profiles along the perturbation axis, while negative correlations indicate concave profiles. Specifically, positive asynchronous correlation intensities manifest as convex curvature in RH-dependent FTIR spectral variations, with negative intensities conversely reflecting concave variation patterns.

A single scan may risk noisy spectra, but the noise is random and high-frequency. For the AS, the IR absorbance peaks (include the sulfate ion, ammonium ion and liquid water) are low-frequency. Due to the spectral resolution of 4 cm⁻¹, the Savitzky-Golay filter with a smoothing width of 21 points was employed to effectively suppress noise while preserving the authentic line shapes of the characteristic infrared absorption bands. A comparative plot of the spectral data before and after processing is presented in Fig.S2, which shows the complete spectrum for raw and smoothed spectrum at 90% (RH). The infrared absorption peaks in the $1500\sim2500$ cm⁻¹ range are primarily attributed to atmospheric water vapor and $CO_2$. To better visualize the spectral variations of sulfate, ammonium, and water, the spectral data are displayed in the $1000\sim1500 cm^{-1}$ and $2500\sim3550 cm^{-1}$, rather than as a complete spectrum. To obtain a credible result, linear baseline corrections and smooth were performed in the regions of $1000\sim1500 cm^{-1}$ and $2500\sim3550 cm^{-1}$ for all infrared spectra before calculations and the analysis. These regions almost cover the absorption features of all identifiable functional groups of interest are selected for analysis. Then we normalize all pre-processed infrared spectra into 2D-IR spectra developed by Kwansei-Gakuin University, Japan (Isao Noda & Ozaki, 2014).

In the PCMW2D correlation analysis, the spectral data is divided by a moving window, with the window size of $N = 2m + 1$. The effect of window size in PCMW2D correlation analysis is studied. The precise window size is a critical factor influencing the results of PCMW2D correlation analysis. Both excessively large and small window sizes adversely affect the results of PCMW2D correlation analysis. In our research, the

window size was set to 2m + 1=7 based on the number of spectra and the perturbation
variable. Significance criteria have been introduced to classify the signals. Specifically,
signals below the 5% threshold are considered noise, while those above the 95%
threshold are regarded as genuine correlation peaks.

**1.2.3 The validation tests for the PCMW2D**

To validate the method, we used it to identify the phase transition point during the
hygroscopic growth of ammonium sulfate (AS) aerosols. Fig.S3(A) shows a
synchronous PCMW2D correlation spectrum constructed from the RH-dependent IR
spectra in the region of $1502\sim1000$ cm$^{-1}$ range during the humidification process of
AS fine particles. From the synchronous correlation spectrum, it can be seen that (1097
cm$^{-1}$, 80%) appears red, which is a positive correlation peak. it indicates that at 80%
humidity, solid sulfate has transformed into aqueous sulfate, and the deliquescence
phase transition point is about 80%. At the same time, ammonium ions also change
most rapidly at this humidity condition (Fig.S3(B)). This suggests that AS may have
undergone deliquescence phase transition at $80\% \pm 0.8\%$ (RH). Which is concordant
with the results derived from Extended Aerosol Inorganics Model (E-AIM)
(http://www.aim.env.uea.ac.uk/aim/aim.php.). Moreover, this result is consistent with
the research results of Ahn et al. (2010)(Ahn et al., 2010), (Y. J. Li, F., C., P., & and
Martin, 2017), who measured the deliquescence point of AS as $79.9\% \pm 0.3\%$ and 79%
$\pm 2\%$ (in Table S1). The hydration characteristics of particles are governed primarily
by their chemical composition, and therefore the deliquescence point of AS is
independent of particle size. Using this approach, we also analyzed the deliquescence
phase transition of 400 nm and 600 nm AS particles during the deliquescence process
and found that both of they were $80\% \pm 0.8\%$ (RH) (Fig.S4). Therefore, this method is
accurate in determining the phase transition point.

## 2. Results and discussion

In this study, the spectra were collected with a 1% (RH) at intervals. For liquid water,
crystal/solid AS, and aqueous AS, their infrared absorption peaks are mainly observed
in 3550-2500 cm$^{-1}$ and 1500-1000 cm$^{-1}$ regions (Onasch et al., 1999; Wei et al., 2022),
(Juan J. Nájera, Percival, & Horn, 2009), (Miñambres, Sánchez, Castaño, &
Basterretxea, 2010). For clarity, the detailed assignments of the infrared spectra bands
are listed in table 1.

**Table 1. The detailed infrared spectra bands assignments of AS aerosol appearing in**

**RH-dependent FTIR**

| Compounds state | species | Peak position Wavenumber/cm$^{-1}$ | Ref |
|---|---|---|---|
| Liquid water | O-H stretching | 3600-3100 | (Onasch et al., 1999; Schlenker & Martin, 2005), (Cai, Luan, Shi, & Zhang, 2017) |
| Crystal/solid AS | NH$_4^+$ deformation ($\nu_4$-NH$_4^+$) | ~ 1417 | (Onasch et al., 1999), (Juan J. Nájera et al., 2009), (Miñambres et al., 2010), (Zhu, Pang, & Zhang, 2022) |
| | SO$_4^{2-}$ stretching | ~ 1084 | |
| Aqueous AS | NH$_4^+$ deformation ($\nu_4$-NH$_4^+$) | ~ 1463 | (Juan J. Nájera et al., 2009), (Miñambres et al., 2010) |
| | SO$_4^{2-}$ stretching | ~ 1097 | |

## 2.1 Humidity-dependent FTIR spectra of AS aerosols upon efflorescence

Fig. 1(A) shows the RH-dependent FTIR spectra of AS in 3600-2550 cm$^{-1}$ region
upon efflorescence. The absorption intensity of the stretching vibration of O–H groups
at ~3400 cm$^{-1}$ decreases, while its peak position remains unchanged during the whole
efflorescence processes. It indicates there is a reduction in liquid water content. When
RH decreased from 90% to 80%, the intensity of the stretching vibration of O–H groups
at ~3400 cm$^{-1}$ decrease by approximately 28%. It can be deduced that it is a gradual
loss of liquid water for the AS solution droplets upon efflorescence. When RH (from
80% to 41%) decreases below the efflorescence point, the intensity of the stretching
vibration of O–H groups at ~3400 cm$^{-1}$ continue decrease by about 42% while aqueous
AS droplets undergo persistent water evaporation and become AS supersaturated state.
These results are consistent with prior studies ((J.-L. Dong et al., 2007; Onasch et al.,
1999). A more pronounced decrease in intensity at ~3400 cm$^{-1}$ occurs as RH falls from
41% to 36%(RH), its intensity ultimately diminishes to zero. It means the breakdown

of hydration networks occurs when almost all condensed-phase water is driven off. AS aerosol would change to fully crystal/solid state at 36% (RH). The efflorescence relative humidity (ERH) occurred between 41% (RH) and 36% (RH), which is consistent with reported values of 34 ～ 40% (RH) in the literature (Cai et al., 2017; Cziczo & Abbatt, 1999; J. J. Nájera & Horn, 2009). It means AS aerosol would be from liquid to supersaturated state, ultimately reaching crystal/solid phase during the efflorescence phase process.

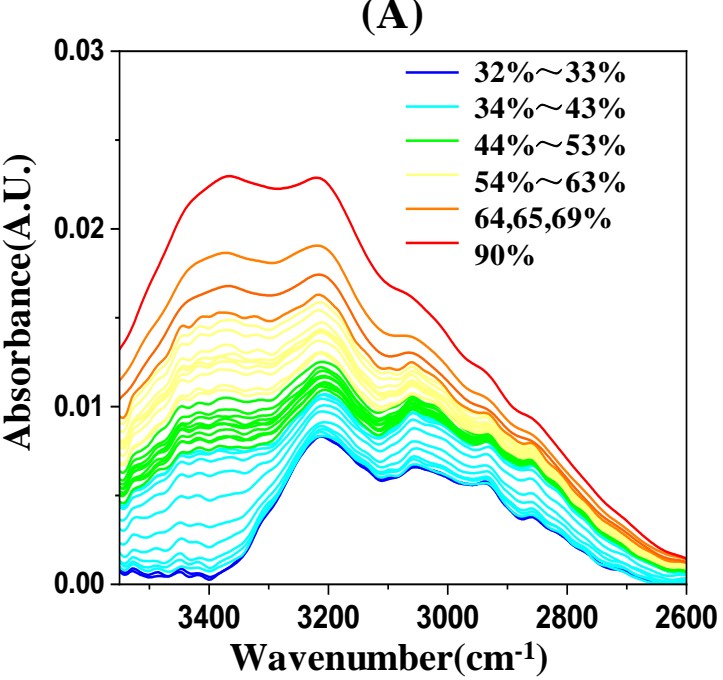

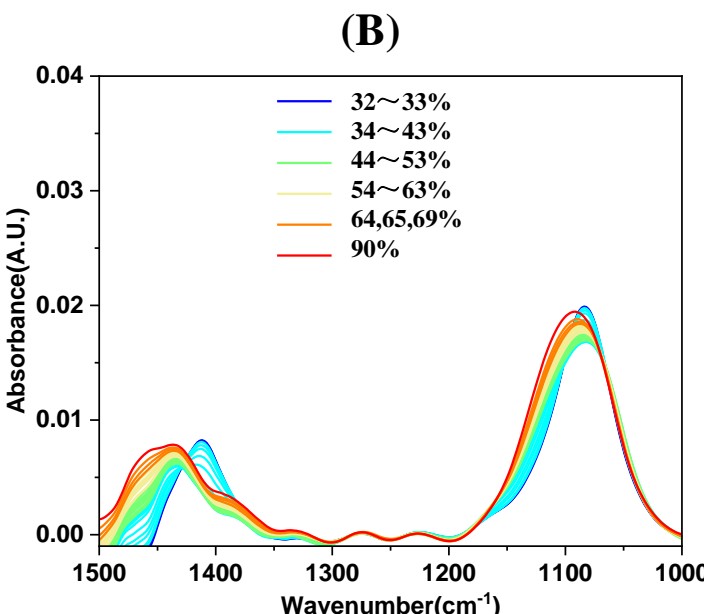

**Fig.1 Humidity-dependent FTIR spectra of AS particles upon efflorescence from 65% to**

**32 at a rate of 1%RH. (A) 3600-2550cm$^{-1}$; (B) 1500-1000cm$^{-1}$.**
$SO_4^{2-}$ acts as a "structure maker" due to its tetrahedral configuration, facilitating
hydrogen bond formation with surrounding water molecules. Similarly, $NH_4^+$
contributes to water structure stabilization through its tetrahedral geometry and
hydrogen bonding capacity (J. L. Dong et al., 2007). The shifting of the $SO_4^{2-}$ and
$NH_4^+$ mode in the FTIR spectra was used to monitor the phase transformation. Fig. 1(B)
shows the RH-dependent FTIR spectra of AS in 1500-1000 cm$^{-1}$ region upon
efflorescence. The $v_3$-$SO_4^{2-}$ bands at 1097 cm$^{-1}$ and 1084 cm$^{-1}$ are characteristic of the
aqueous and crystalline states, respectively; the areas of these absorption peaks are
proportional to the relative mass/concentration of sulfate ions present in each state.
Similarly, the $v_4$-$NH_4^+$ bands at 1463 cm$^{-1}$ and 1417 cm$^{-1}$ correspond to the aqueous
and crystalline states, respectively, with their peak areas likewise correlating with the
relative mass/concentration of ammonium ions in each state. As RH decreased from
90% to 41%, both of the absorption peaks areas/intensities at 1094 cm$^{-1}$ and 1463 cm$^{-1}$
gradually diminished, indicating a continuous decline in the relative
mass/concentration of aqueous sulfate and ammonium ions. Which is a direct
consequence of progressive water loss from the droplet solution. When RH decreased
from 41% to 36%, a distinct phase transition occurred between 41% (RH) and 36%
(RH), characterized by an abrupt red-shift in the $v_4$-$NH_4^+$ peak position from 1463 cm$^{-1}$
$^{-1}$ to 1417 cm$^{-1}$, accompanied by a sharp intensity increase, signaling the onset of
anhydrous crystal formation. Similarly, an abrupt red-shift in the $v_3$-$SO_4^{2-}$ band from
1097 cm$^{-1}$ to 1084 cm$^{-1}$, accompanied by a sharp intensity increase, was observed
during crystallization. Notably, the observed red-shift of the $SO_4^{2-}$ stretching band can
be explained by the stronger hydrogen bonding between $SO_4^{2-}$ and $NH_4^+$ compared to
that between $SO_4^{2-}$ and $H_2O$. As $H_2O$ molecules in the hydration shell are replaced by
$NH_4^+$ ions during crystallization, the effective mass of the vibrating oxygen atoms
increases, leading to the observed frequency decrease—consistent with the mechanism
discussed in prior studies(J.-L. Dong et al., 2007). Complete crystallization was
achieved at 36% (RH) when the intensities at 1084 cm$^{-1}$ and 1417 cm$^{-1}$ were constant.
The complete replacement of water molecules by ammonium ions in the hydration shell
of sulfate ions indicated the completion of the crystallization process. So the
crystallization threshold of AS was identified at 41% (RH) and completed at 36%(RH),
consistent with prior studies (Cai et al., 2017; Cziczo & Abbatt, 1999; J. J. Nájera &
Horn, 2009) and further confirmed by the results of the stretching vibration of O–H

groups (at ~3400 cm⁻¹). These variations in the peaks with RH are consistent with the convex/concave variations (Fig.2(A) and Fig.2(B)). A similar trend was observed in the O-H stretching vibration modes, where absorption peak intensities exhibited analogous humidity-dependent variations for ammonium and sulfate ions in aqueous AS also gradually decreased. But these absorption peaks of ammonium and sulfate ions do not disappear but underwent a red shift(Zhu et al., 2022). The reason is that when AS transitions from liquid to crystal/solid state, the local environment of the ammonium and sulfate ions change.

## 2.2 PCMW2D Correlation Analysis of AS aerosols upon efflorescence

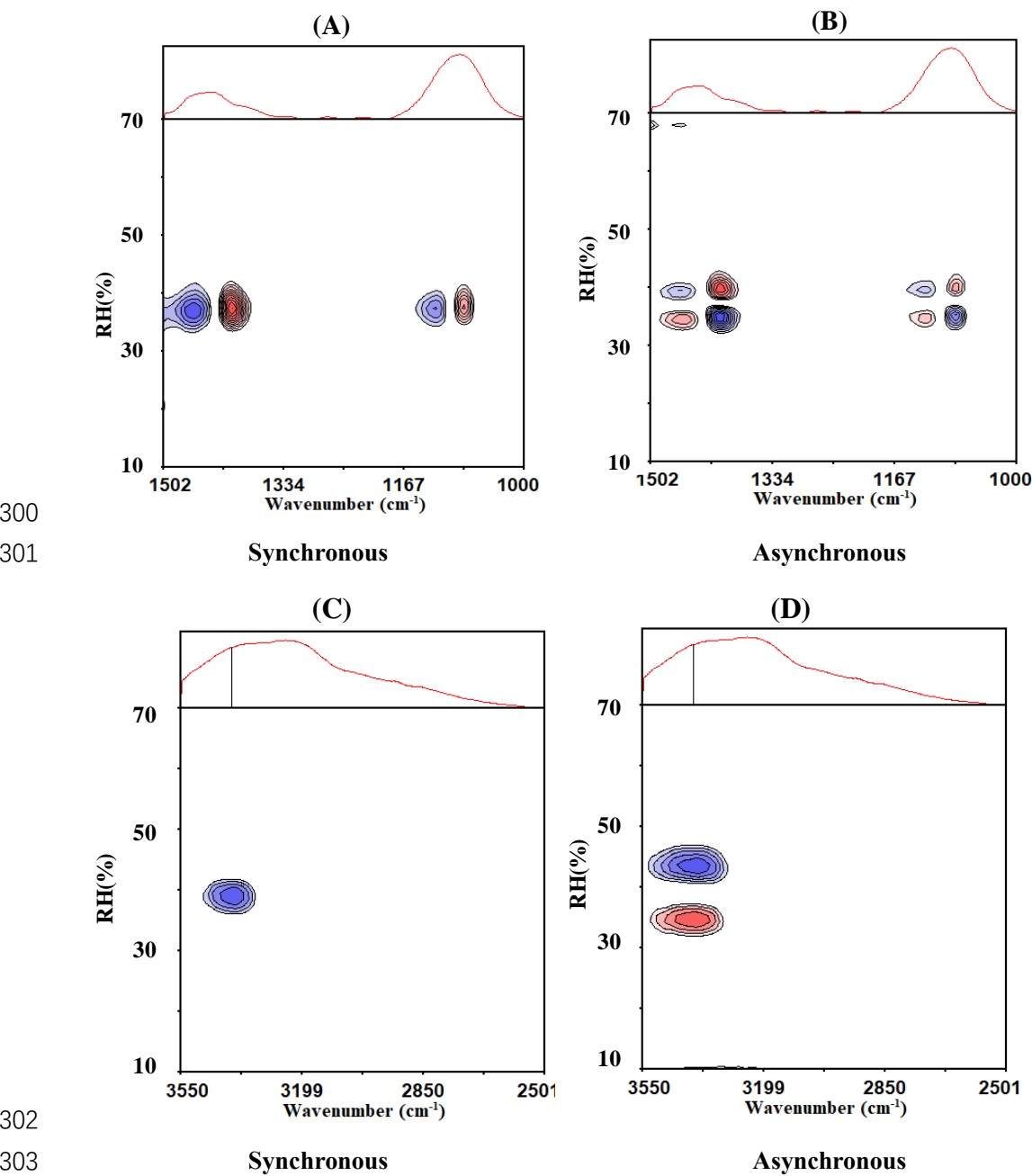

**Fig.2 (A)Synchronous and (B)asynchronous PCMW2D spectra in the 1502-1000 cm$^{-1}$ region during the efflorescence process of AS aerosol; (C)Synchronous and (D)asynchronous PCMW2D spectra in the 3550-2501 cm$^{-1}$ region during the efflorescence process of AS aerosol. Red and blue means a positive and negative correlation value, respectively.**

To accurately determine the ERH (efflorescence relative humidity) of AS, we further analysis the PCMW2D spectra, as they can determine the phase transition points and regions. Fig.2(A) gives the synchronous PCMW2D spectra in 1502-1000 cm$^{-1}$ region during the efflorescence process of AS aerosol. As can be seen, (1084 cm$^{-1}$, 39%) and (1417 cm$^{-1}$, 39%) show red color, which is a positive correlation peak, indicating that crystal/solid sulfate and ammonium ions concentration increased at 39%(RH). While (1097 cm$^{-1}$, 39%) and (1463 cm$^{-1}$, 39%) are negative correlation peaks, indicating that aqueous sulfate and ammonium ions concentration diminished. Combined with one-dimensional FTIR spectroscopy (Fig.1), which shows the transformation of aqueous ammonium and sulfate ions to the solid state at 39% (RH), these spectral features collectively provide clear evidence that the phase transition occurs at this RH. This means the ERH of AS aerosols is 39% ± 0.8% (RH). It is consistent with the findings of Takahama et al. (2007)(Takahama, Pathak, & Pandis, 2007), Yeung et al. (2009) (Yeung, Lee, & Chan, 2009), who measured the deliquescence point of AS at 38 ~ 40% (RH) (Xu, Imre, McGraw, & Tang, 1998). Therefore, this method can be used to analyze the aerosol phase transition process.

Fig.2(B) is the same as (A) but the asynchronous PCMW2D spectra, which allow for the identification of pronounced spectral intensity variation regions. (1084cm$^{-1}$, 41%) shows red as a positive correlation peak, and (1084cm$^{-1}$, 36%) shows blue as a negative correlation peak. It demonstrated that the 1084 cm$^{-1}$ band intensity rose convexly and concavely by 41% (RH) and 36% (RH), respectively. Conversely, (1097cm$^{-1}$, 36%) showed red, which was a positive correlation peak, and (1097cm$^{-1}$, 41%) showed blue, which was a negative correlation peak. It indicated the 1097 cm$^{-1}$ band intensity concavely and convexly decreases at 41% and 36%. It also indicates the transformation of sulfate from an aqueous phase to a solid AS aerosol. similarly, ammonium ions exhibit the same phenomenon. Regarding the ammonium ion, the following demonstrates its transformation. (1417cm$^{-1}$, 41%) showed red, which was a

positive correlation peak, and (1417cm$^{-1}$, 36%) showed blue, which was a negative
correlation peak. (1463cm$^{-1}$, 36%) showed red, which was a positive correlation, and
(1463cm$^{-1}$, 41%) showed blue, which was a negative correlation.
Based on the quantitative analysis of the synchronous and asynchronous
correlation maps, it can be concluded that during the transformation of supersaturated
ammonium sulfate (AS) from the liquid to the solid state, the phase transition point
was determined to be 39% (RH), with a transition region spanning from 36% to 41%
(RH). The identified phase transition range (RH: 36～41%) shows remarkable
consistency with the reported deliquescence behavior of AS (Xu et al., 1998). Fig1.(A)
and Fig.1(B) illustrate the variation of these absorption peaks with RH.
In order to clearly analyze the variation of infrared absorption peaks at 1084 cm$^{-1}$
$^{1}$, 1097 cm$^{-1}$, 1417 cm$^{-1}$ and 1463 cm$^{-1}$, we listed the two-dimensional correlated
infrared absorption peaks in Fig.2 and gave them in Table 2. Based on the PCMW2D
reading rules, the signs and intensity profiles of the synchronous and asynchronous
correlation peaks provide direct evidence for the phase transition
mechanism. Specifically, aqueous sulfate and ammonium ion concentration convex
decrement at 41% (RH), linear decrement 39% (RH), at last concave decrement at 36%
(RH). And the crystal/solid sulfate and ammonium show the opposite trend. The multi-
stage efflorescence mechanism of AS, which was quantitatively resolved from the
correlation spectra, aligns closely with the efflorescence relative humidity (ERH: 38～
40% RH) reported in previous studies. This agreement validates the capability of
PCMW2D analysis in providing data-driven insights into aerosol phase transitions.
**Table 2. The positions and symbols of the peaks, the transition RH determined upon**
**efflorescence from Fig. 2**

| Synchronous | Asynchronous | Spectra change | |
|---|---|---|---|
| (1084cm$^{-1}$, 39%) + | (1084cm$^{-1}$, 36%)- | ↱ | 36↱39↗41 |
| | (1084cm$^{-1}$, 41%) + | ↰ | |
| (1097cm$^{-1}$, 39%) - | (1097cm$^{-1}$, 36%) + | ↘ | 41↘39↘36 |
| | (1097cm$^{-1}$, 41%) - | ↳ | |
| (1417cm$^{-1}$, 39%) + | (1417cm$^{-1}$, 36%) - | ↱ | 36↱39↗41 |
| | (1417cm$^{-1}$, 41%) + | ↰ | |
| (1463cm$^{-1}$, 39%) - | (1463cm$^{-1}$, 36%) + | ↘ | 41↘39↘36 |
| | (1463cm$^{-1}$, 41%) - | ↳ | |

Fig.2(C) and (D) are the same as (A) and (B) but in the region of 3550-2501cm$^{-1}$. From the synchronous correlation spectrum, it can be seen that (3400 cm$^{-1}$, 39%) appears blue, which is a negative correlation peak. From the asynchronous correlation spectrum, it can be seen that (3400 cm$^{-1}$, 41%) and (3400cm$^{-1}$, 36%) appears blue and red, respectively. There is a negative and positive correlation peak, respectively. Therefore, we can infer that the efflorescence phase transition point was 39% (RH). The aqueous AS began to lose water at 41% (RH), and crystallization occurred at 39% (RH). At last, AS particles completely turned into crystal/solid state and the condensed water has completely disappeared at 36% (RH).

Therefore, AS exists in a supersaturated state when the RH is above 41% (J. L. Dong et al., 2007). AS the RH decreases from 41% to 39%, the liquid water molecules around the aqueous sulfate ions and ammonium ions begin to evaporate slowly. Pronounced efflorescence occurs at 39% (RH). When the RH further drops to 36%, the liquid water molecules around the sulfate ions are completely replaced by ammonium ions, and the AS particles transition from liquid state to a crystal/solid state. So the efflorescence phase transformation process completed.

## 2.3 Generalized 2D-IR correlation spectra analysis of AS aerosols upon efflorescence

**(A)**                    **(B)**

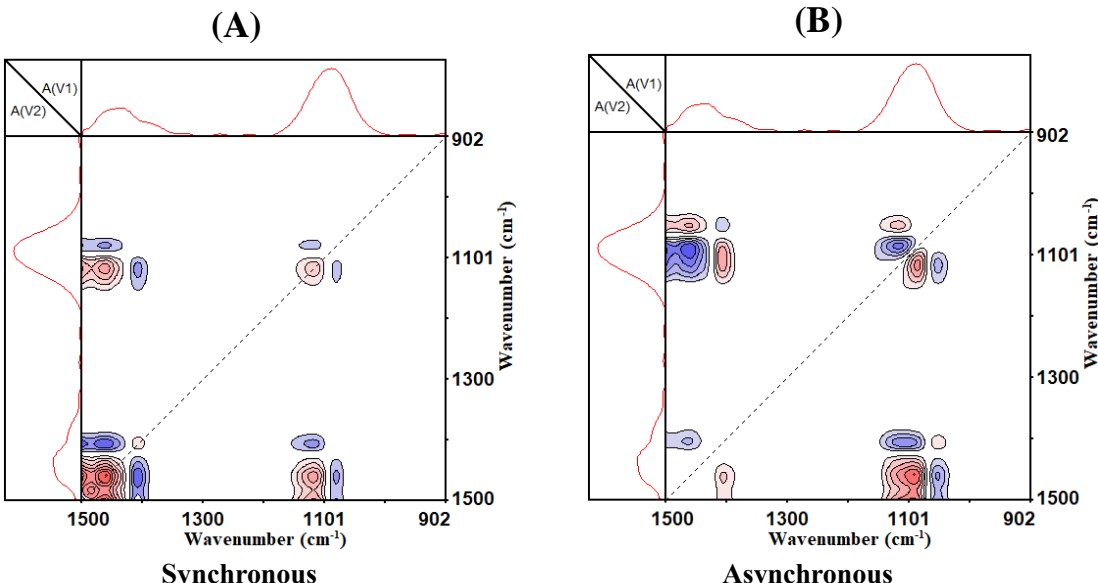

Synchronous                    Asynchronous

**Fig. 3(A) synchronous and (B) asynchronous 2D-IR correlation spectra of AS aerosol in the 1500-902 cm-1 region upon efflorescence. Red and blue color represent positive and negative correlations, respectively. RH ranges is 90～20% (RH), red and blue means a positive and negative correlation value, respectively.**

Fig.3 shows the generalized 2D-IR correlation spectra of AS particles upon efflorescence, which provides enhanced spectral resolution and reveals the sequential order of peak intensity changes. In the synchronous 2D-IR correlation spectra, five strong auto-peaks are observed at (1097, 1463) $cm^{-1}$, (1097, 1084) $cm^{-1}$, (1417, 1463) $cm^{-1}$, (1097, 1417) $cm^{-1}$, and (1084, 1463) $cm^{-1}$, indicating significant changes in these infrared absorption bands during the efflorescence phase transition. The positive cross-peak at (1097, 1463) $cm^{-1}$ demonstrates synchronized intensity increases, indicating a simultaneous decrease in the concentration of aqueous sulfate and ammonium ions. Four negative cross-peaks were observed at (1097, 1084) $cm^{-1}$, (1417, 1463) $cm^{-1}$, and (1097, 1417) $cm^{-1}$, (1084, 1463) $cm^{-1}$. The negative cross-peak at (1097, 1463) $cm^{-1}$, demonstrates opposite changes of the spectral intensities, indicating a decrease in aqueous sulfate ions and an increase in solid sulfate ions, suggesting a transition of aqueous sulfate ions to the solid phase. Similarly, the negative cross-peak at (1417, 1463) $cm^{-1}$ demonstrates a transition of aqueous ammonium ions to the solid phase. So aqueous sulfate and ammonium ions would change to crystal/solid sulfate and ammonium ions. It would result in the increase of crystal/solid sulfate and ammonium ions concentrations upon efflorescence.

In the asynchronous 2D-IR correlation spectra, seven strong auto-peaks are present. Among them: (1417, 1463) cm$^{-1}$, (1097, 1463) cm$^{-1}$, (1084, 1417) cm$^{-1}$, and (1097, 1084) cm$^{-1}$ exhibit positive auto-peaks. While (1097, 1417) cm$^{-1}$, (1084, 1463) cm$^{-1}$, and (1084, 1097) cm$^{-1}$ show negative auto-peaks. Since asynchronous spectra reflect differing rates of spectral changes, the sequence of molecular bond transformations during efflorescence can be deduced as: The spectral variations at 1084 cm$^{-1}$ appear earlier in the humidity-driven process than those at 1097 cm$^{-1}$, which in turn precede the changes at 1463 cm$^{-1}$, and finally, the variations at 1417 cm$^{-1}$ occur last.

By correlating the absorption band positions of $SO_4^{2-}$ and $NH_4^+$ in different states with the 2D-IR transition sequence, we infer that the concentrations increase in crystal/solid sulfate ions and decrease in aqueous sulfate ions do not occur simultaneously. Instead, the efflorescence mechanism processes in two main distinct steps: at initial stage, crystal/solid sulfate ions concentration rises as liquid water is lost from the supersaturated AS particles. Then at secondary stage, the replacement of $H_2O$ by $NH_4^+$ in the sulfate hydration shell, evidenced by a distinct spectral red-shift, entails a bond reorganization in which stronger $SO_4^{2-}$–$NH_4^+$ bonds supersede the original sulfate-water interactions. Eventually, efflorescence occurs completely, forming crystal/solid AS.

## 2.4 The micro-dynamic mechanism during efflorescence process

The two-dimensional correlation infrared spectroscopy results further confirm at the molecular level that the microscopic dynamics of AS aerosols during the phase transition process. During the efflorescence process, the formation of crystal/solid sulfate ions initiates the AS efflorescence mechanism. Specifically, the crystal/solid sulfate ions undergoes an increase pattern of first slow, then fast, and finally slow again, while the liquid sulfate ions exhibit a decrease pattern of first fast, then slow, and then fast again. Following this, the crystal/solid ammonium ions experience a similar slow-fast-slow increase trend, while the liquid ammonium ions decrease. Based on the observed spectral sequences, we propose a potential mechanism in which the efflorescence process may involve the initial reorganization of sulfate ions followed by ammonium ions, ultimately leading to the complete dehydration of the particles and the formation of crystalline AS. Given the established surface preference of anions (Chamberlayne & Zare, 2020), we infer that this process in atmospheric AS aerosols is

primarily driven by surface-induced nucleation. which aligns with the findings of Ciobanu et al.(Ciobanu et al., 2010). Thus, as RH decreases, the replacement of liquid water by ammonium ions around the sulfate ions leads to the formation of crystal/solid sulfate ions. Based on the two-dimensional correlation infrared spectroscopy (2D-COS) analysis of ammonium sulfate (AS) aerosols, the data are consistent with a multi-step efflorescence mechanism of AS. A schematic representation of this phase transition process is provided in Fig. 4.

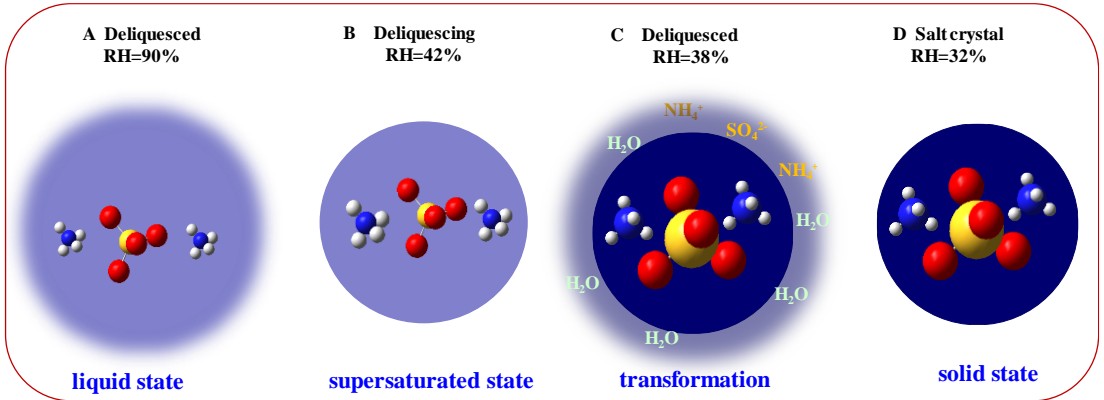

**Fig. 4 Schematic view of the micro-dynamic mechanism during efflorescence process**

The efflorescence process is hypothesized to begin with liquid water loss from the supersaturated liquid surface of AS aerosols, potentially leading to the initial formation of crystalline sulfate ions. Subsequently, as more water molecules surrounding in the hydration shell of the aqueous sulfate ions are displaced by ammonium ions, the sulfate ions transition into the crystal/solid phase, accompanied by a gradual blue shift in the sulfate absorption peak. Concurrently the gradual dehydration of ammonium ions in the bulk phase may facilitate their approach toward sulfate ions, forming crystal/solid AS. As humidity further decreases, the remaining liquid water molecules eventually detach from the AS system, completing the liquid-to-solid phase transition. Thus, the entire transformation process from liquid AS to crystal/solid AS is driven by sequential dehydration, ion reconfiguration, and crystallization.

## 3. Conclusions

In this work, a precision determination methodology was developed to analysis the phase transition mechanism of aerosols by coupling a RH controlling system and 2D-IR. We measured the phase transition efflorescence point at 39% ± 0.8% (RH). Which confirms that this method is accurate in determining the phase transition point.

By correlating the absorption band positions of sulfate in different states with the 2D-COS-derived transition sequence, we infer that the increasement in

crystal/solid sulfate and decrease in liquid sulfate ions do not occur simultaneously. Instead, the efflorescence process involves four steps:(1) At initial stage (RH: from 90% to 80%):When RH decreased from 90% to 80%, the intensities of these peaks decrease. It can be deduced that this tendence upon efflorescence can be explained by a gradual loss of liquid water for the solution droplets. (2) At second stage (RH: from 80% to 41%): When RH decreased from 80% to 41%, aqueous AS droplets undergo persistent water evaporation and become AS supersaturated state. (3) At third stage (RH: from 41% to 39%): When RH decreased from 41% to 39%, as more water molecules from the supersaturated AS particles are gradually lost, the sulfate ions transition into the crystal/solid phase. The replacement of $H_2O$ by $NH_4^+$ in the sulfate hydration shell, evidenced by a distinct spectral red-shift, entails a bond reorganization in which stronger $SO_4^{2-}$–$NH_4^+$ bonds supersede the original sulfate-water interactions. Efflorescence occurs and forms crystal/solid AS. (4) At last stage (RH: from 39% to 36%): As humidity further decreases, the remaining liquid water molecules eventually detach from the AS system, completing the liquid-to-solid phase transition. Thus, the entire transformation process from liquid AS to crystal/solid AS is driven by sequential dehydration, ion reconfiguration, and crystallization.

While the efflorescence properties in this study are consistent with earlier reports, the application of 2D-IR spectroscopy has enabled the elucidation of more sophisticated structural evolution patterns and the precise sequence of hydrogen-bonding rearrangements among $NH_4^+$, $SO_4^{2-}$, and $H_2O$. These findings deepen the mechanistic understanding of aerosol phase transitions and nucleation, which could advance predictive models for inhaled particle behavior. Additionally, the distinct bonding characteristics observed in AS droplets may offer essential foundational data in atmospheric heterogeneous chemistry.

## Data availability

The data shown in the paper is available on request from the corresponding authors.

## Author contributions

XW designed the experiment, carried out the data analysis and wrote the paper with contributions from all co-authors; HG contributed to scientific discussions; XL contributed to this work by providing formal analysis, JW, DW and JL contributed to this work by providing constructive comments.

## Competing interests

The contact author has declared that none of the authors has any competing interests.

## Disclaimer

Publisher's note: Copernicus Publications remains neutral with regard to jurisdictional claims made in the text, published maps, institutional affiliations, or any other geographical representation in this paper. While Copernicus Publications makes every effort to include appropriate place names, the final responsibility lies with the authors

## Acknowledgments

This work was financially supported by the National Natural Science Foundation of China (Nos. U2133212, 42375124), National Key Research and Development Program of China (No.2023YFC3705405), the Natural Science Foundation of Anhui Province (No. 2108085MD139).

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
