# Peer review of "Identification of Micro-dynamics Phase Transition Processes for"

_EGUsphere, 2025_

## Author Comment (AC1)

**Response to Referee #1:**

Thanks very much for your comments, suggestions and recommendation with respect to improve this paper. The response to all your comments is listed below.

The manuscript entitled "Identification of Micro-dynamics Phase Transition processes for Ammonium Sulfate aerosols by Two-dimensional Correlation Spectroscopy" investigates the micro-dynamic mechanisms of ammonium sulfate (AS) aerosol phase transitions using two-dimensional correlation infrared spectroscopy (2D-IR), coupled with relative humidity (RH) control. By employing 2D-IR (including generalized 2D-IR and perturbation-correlation moving window 2D (PCMW2D) spectroscopy), the authors successfully elucidate non-equilibrium micro-dynamic processes during AS efflorescence, revealing four distinct sequential steps at the molecular level. This approach advances beyond conventional methods (e.g., ESEM, H-TDMA) that primarily characterize physical parameters (size, shape), offering unprecedented insights into intermolecular interactions (e.g., hydrogen bond dissociation, ion reconfiguration). The manuscript may be suitable for publication after major revisions and addressing the following concerns.

1. The statement "This study could provide critical insight about redefining atmospheric heterogeneous chemistry." may be not suitable since little information was provided regarding heterogeneous atmospheric chemistry.

**Response:** We have followed your suggestion and don't emphasize this point. We have changed some sentences in the introduction. Please check the marked up file for details.

2. How to ensure the accuracy of a 1% RH interval in experiments that is close to the error of a hygrometer?

**Response:** The FTIR spectrometer starts to take absorption spectra of the samples approximately 1 min after the injection of each designated RH. This time interval is used to stabilize the RH inside the sample cell. We measured the RH at the outlet of the cell and confirmed that the difference from the inlet RH did not exceed 0.5% (RH). Moreover, we use a humidity detector to continuously monitor changes. Here, the 1%(RH) refers to a relative change in reading, not an absolute value. Although the detector has an accuracy of  $\pm 0.8\%$ , its systematic error remains consistent between the

two humidity points and is largely canceled out when calculating the difference. Therefore, a 1% (RH) change interval is achievable. Please check the marked up file for details.

3. The FTIR measurement is reported to use "a repeat time of 1 scan". This is problematic because FTIR spectroscopy typically requires multiple scans (e.g., 32 or 64) to average out noise and improve SNR. A single scan risks capturing unreliable, noisy spectra, which could distort the identification of subtle spectral changes during phase transitions and may introduce artifacts in the synchronous/asynchronous 2D-IR maps, affecting the precision of transition point determinations.

Response: Multiple scans could average out noise and improve SNR, but it need more time (about 2~3min). A single scan may risk noisy spectra, but the noise is random and high-frequency. And for the AS, the IR absorbance peaks (include the sulfate ion, ammonium ion and liquid water) are low-frequency. A single scan could be smoothed to improve SNR. So the linear baseline corrections and smooth were performed in the regions of 1000~1500cm-1 and 2500~3550cm-1 for all infrared spectra before calculation and analysis. Then we normalize all pre-processed infrared spectra into 2D-IR spectra. From Fig.1, the pre-processed infrared spectra clearly show the absorption peaks of sulfate, nitrate, and ammonium ions, and allows observation of their peak shifts with changes in relative humidity (RH). And the calculation results in the synchronous/asynchronous 2D-IR maps (Fig.2) were also consistent with the observations reported by Takahama et al. (2007)1, Yeung et al. (2009) 2. So a single scan could meet the requirements of the transition point determinations precision. Moreover its precision was not be affected. We have included this interpretation in the revised version. Please check the marked up file for details.

4. How is the sample particle size determined to be 300nm?

**Response:** In our paper, we used a differential mobility analyzer (DMA) to sort the diameter of the particles. The diameter of 300 nm in the experiment description section should also be electrical mobility diameter. Yan et al. (2020) have compared the  $D_{ve}$  and the  $D_{em}$  of AS sorted by the identical DMA of this study, which is shown in the figure R1. A good agreement between  $D_{em}$  and  $D_{ve}$  for ~100 nm AS was observed by Yan

et al. (2020). For the AS with an electrical mobility diameter of  $\sim$ 100 nm, its volume equivalent diameter is  $\sim$  94 nm. Please check the marked up file for details.

Figure R1 The SEM image of 100 nm AS particles deposited on the silicon wafer.

5. The PCMW2D and generalized 2D-IR analyses are central to identifying transition points and molecular sequences, but their reliability is compromised by incomplete data. For instance, the assignment of positive/negative correlation peaks in Fig. 2, references wavenumbers (e.g., 1084 cm-1, 1417 cm-1) that lack contextual spectral data (e.g., how these peaks evolve with RH). Without clear baseline spectra or RH-dependent intensity curves, the interpretation of "convex/concave variations", remains speculative.

**Response:** Fig. 1(B) shows the RH-dependent FTIR spectra of AS in 1500-1000 cm-1 region upon efflorescence. The blue arrows specifically show the infrared absorption peaks at 1463 cm-1 and 1097 cm-1, respectively. The blue downward arrows indicate a decrease in absorption peak intensity with decreasing RH. The red arrows specifically show the infrared absorption peaks at 1417 cm-1 and 1084 cm-1, respectively. The red upward arrows indicate an increase in absorption peaks intensity with decreasing RH. It can be observed that as the humidity decreases, the absorption peak intensities at 1463cm-1 and 1097 cm-1 gradually weaken. At 39% (RH), the 1463 cm-1 and 1097 cm-1 absorption peaks shift to 1417 cm-1 and 1084 cm-1, respectively.

Fig.2(A) gives the synchronous PCMW2D spectra in  $1502 \sim 1000$  cm-1 region during the efflorescence process of AS aerosol. As can be seen, (1084 cm-1, 39%) and (1417 cm-1, 39%) show red color, which is a positive correlation peak. While (1097 cm-1, 39%) and (1463 cm-1, 39%) are negative correlation peaks.

So these variations in the peaks (Fig. 1(B)) with RH are consistent with the convex/concave variations (Fig.2(A) and Fig.2(B)). And these variations in the peaks (Fig. 1(A)) with RH are consistent with the convex/concave variations (Fig.2(C) and Fig.2(D)). We have included this interpretation in the revised version. Please check the marked up file for details.

Fig.1 Humidity-dependent FTIR spectra of AS particles upon efflorescence from 65% to 32% at a rate of 1%RH. (B) 1500-1000cm-1. The blue downward arrows indicate a decrease in absorption peak intensity with decreasing RH. And the red upward arrows indicate an increase in absorption peaks intensity with decreasing RH.

6. Is the attribution of infrared peaks based on references? Usually, NH4+ exhibits several infrared absorption peaks around 3000 cm-1. Why are the NH4+ peaks of the dried sample not obvious around 3000 cm-1 in this study?

**Response:** Fig.R2 the fitted bands for spectra in the region of 2600~3800 cm-1 at 90% (RH) which show three infrared absorption peaks for NH4+ and one infrared absorption peak for liquid water. Fig.R3 show the fitted bands for spectra in the region of 2600~3400 cm-1 at 34% (RH) which show NH4+ exhibits three infrared absorption peaks. There are 2887 cm-1, 3036 cm-1, 3209 cm-1 for the dried sample at 34% (RH), which is similar with the results of Onasch et (in Fig. R4) 3, Juan et 4. We have included this interpretation in the revised SI version. Please check the marked up file for details.

Fig.R2 the fitted bands for spectra in the region of  $2600 \sim 3800 \text{ cm}^{-1}$  at 90% (RH). Dark line: the experimental IR absorbance spectra. Blue and red lines are NH4+ and liquid water fitted spectral bands, respectively. Magenta lines: the sum spectra of the fitted spectral components.

Fig.R3 the fitted bands for spectra in the region of  $2600 \sim 3400 \text{ cm}^{-1}$  at 34% (RH). Dark line: the experimental IR absorbance spectra. Blue lines are NH4+ fitted spectral bands, respectively. Magenta lines: the sum spectra of the fitted spectral bands.

Fig.R4 the IR spectra for AS in different RH3. (A) the initial dry AS aerosol;(B) the initial uptake of liquid water by AS aerosol; (C) totally deliquesced aerosol sample; (D) aerosol on the water uptake curve above the deliquescence point.

7. How to define complete dryness in experiments? From the infrared spectrum, it appears that there is still a significant amount of H2O present in the sample.

**Response:** This effect was caused by background subtraction, and we have reprocessed the background correction. Please check the marked up file for details.

8. There are many spelling and grammar errors. The issue of capitalization of the first letter. For example, Line 53, Line 57, Line 173, these sentences seem not complete.

**Response:** Done. Please check the marked up file for details.

9. Line 218: DRH

**Response**: Done. Please check the marked up file for details.

**Reference:**

- 1. Takahama, S.; Pathak, R. K.; Pandis, S. N., Efflorescence Transitions of Ammonium Sulfate Particles Coated with Secondary Organic Aerosol. *Environmental Science & Technology* **2007**, *41* (7), 2289-2295.
- 2. Yeung, M. C.; Lee, A. K. Y.; Chan, C. K., Phase Transition and Hygroscopic Properties of Internally Mixed Ammonium Sulfate and Adipic Acid (AS-AA) Particles by Optical Microscopic Imaging and Raman Spectroscopy. *Aerosol Science and Technology* **2009**, *43* (5), 387-399.
- 3. Onasch, T. B.; Siefert, R. L.; Brooks, S. D.; Prenni, A. J.; Murray, B.; Wilson, M. A.; Tolbert, M. A., Infrared spectroscopic study of the deliquescence and efflorescence of ammonium sulfate aerosol as a function of temperature. *Journal of Geophysical Research: Atmospheres* **1999**, *104* (D17), 21317-21326.

4. Nájera, J. J.; Horn, A. B., Infrared spectroscopic study of the effect of oleic acid on the deliquescence behaviour of ammonium sulfate aerosol particles. *Physical Chemistry Chemical Physics* **2009**, *11* (3), 483-494.

---

## Author Comment (AC2)

**Response to Referee #2:**

Thanks very much for your comments, suggestions and recommendation with respect to improve this paper. The response to all your comments is listed below.

The paper proposes a new method for investigating ammonium sulfate (AS) efflorescence using two-dimensional correlation infrared spectroscopy (2D-IR). Coupling this method with FTIR and PCMW2D, the authors attempt to validate the microdynamic mechanisms behind AS efflorescence. Although the method has potential and the topic is timely, the manuscript in its current form falls short of the standards expected for AMT, particularly in terms of methodological detail, validation, and clarity of interpretation. I therefore recommend rejection at this stage. With substantial additional work and restructuring, however, I believe this research could form the basis of a stronger future submission.

**Response:** All your comments listed below have been addressed. Please check the point-by-point response as follows.

First, I want to say that I initially was happy to receive this review, as it is a topic in which I want to deepen my knowledge. The authors would excuse me for the long time I took to properly review their manuscript as I first wanted to be fully familiar with literature. After reading the introduction, I noted the absence of several key references on efflorescence, which makes it difficult for a reader to appreciate the state of the art and the potential contribution of 2D-IR. Including these references and clarifying the recent advances would greatly strengthen the paper. Importantly, because of several vague sentences, the paper often posits 'old' state of the art as traditional understanding, without clearly explaining the more recent results. This substantially weakens the introduction. Unlike what is stated, the paper does not provide a kinetic mechanism of AS efflorescence which has been proposed through literature (Onasch 1999, Takahama 2007, Ciobanu 2010, Wang 2017, Xu 2022). This should be clearly stated in introduction. Several sentences are overstated and should be revised for accuracy. For instance: "The traditional understanding assumes that phase transitions are thermodynamically equilibrium," (L15), "the molecular-scale dynamical processes governing aerosol phase transitions remain hitherto uncharacterized" (L77) should be revised to reflect more accurately the current understanding in the literature. Thus, I would recommend to have an introduction that is in 3 to 4 paragraphs: the first one, presenting the importance of understanding aerosol transition, the 2nd one on equilibrium and kinetic mechanisms that are currently known, the 3rd would present the methods that arrive to these conclusions and highlight the current limitation, and a 4th one could introduce 2D-IR as a promising method that needs to be investigated and validated as done in the paper.

**Response:** Done. We have followed your suggestion and revised the introduction. A kinetic mechanism of AS efflorescence has been stated in introduction and some overstated sentences have been revised.

In my opinion, the manuscript oscillates between two positions. Either it is a "method paper", which aims to demonstrate the value and robustness of an approach (in this case 2D-COS applied to aerosols). Or it is a "process paper", which seeks to provide new scientific insights into the efflorescence of AS. However, in its current form, it mixes the two approaches without clarifying its angle. Many of the results presented are already known generalities (efflorescence values, qualitative sequence of the process), while the methodological contribution is not sufficiently validated (next point). I recommend that the authors clearly refocus their contribution. In AMT, it seems to me that the methodological contribution is the most relevant: it should therefore be highlighted, demonstrating the reproducibility, validation and advantages of the method compared to existing approaches, rather than claiming to offer a scientific (re)interpretation of efflorescence.

Accordingly, we would expect from a "method paper" to read many details on the method, which currently lacks. In its current format, the paper lacks sufficient detail for reproducibility. To make the method accessible to readers, I suggest considering the following points. But I want to stress that this list is probably not exhaustive, and any detail the authors think would be necessary should be integrated.

1. The paper refers to Wei et al., 2022 for the description of the experimental setup. Where elements of the setup and analysis are literally identical to Wei et al. (2022), a concise reference is acceptable (e.g., 'atomizer/dryer/DMA layout as in Wei 2022'). However, parameters that are specific to the present study must be reported explicitly here, as they directly affect FTIR intensities, 2D-COS outcomes, and the inferred efflorescence thresholds. For instance, we need details on the 300 nm selection and its distribution, mass/areal loading on ZnSe (and resulting optical thickness),

deposition morphology and homogeneity, the exact RH program and dwell strategy during dehumidification.

**Response:** The mass of the deposited AS is about 0.074 mg which was measured by Ion chromatograms (ICS-3000, Dionex, United States) and the area loading on ZnSe is about 0.79cm2. Fig.S1shows the ion chromatograms of the deposited AS. Since this area is smaller than the infrared spot illuminating the sample, the acquired infrared spectrum represents the total absorption spectrum of AS. There were about billion particles deposited onto the substrate. The deposited particulate film had a thickness equivalent to approximately three particle monolayers. The measured particles deposited onto the substrate could overlap on each other for FTIR. But the chemical composition is not changed for the 300nm particles during depositing onto the stacking state. And for the 300nm particles, its hydration characteristic mainly depends on its chemical composition and the kelvin effect is negligible. The enrichment for the particles is to improve the signal of FTIR because the dehumidifying signal of a single particle is too weak to be measured by the FTIR method. Since the chemical composition is not changed for the 300nm particles during depositing onto the stacking state, the concentration of the particles deposited does not influence our results. The FTIR spectrometer starts to take absorption spectra of the samples approximately 1 min after the injection of each designated RH. This time interval is used to stabilize the RH inside the sample cell. We also measure the RH at the outlet of the cell and found that it is close to that at the inlet one. We have included this explanation in the revised paper.

Fig.S1 Ion chromatograms of the deposited AS. Ion chromatograms (ICS-3000, Dionex, United States)

For the AS with an electrical mobility diameter of  $\sim 100$  nm, its volume equivalent diameter is  $\sim 94$  nm. Please check the marked up file for details.

Fig. R1 The SEM image of 100 nm AS particles deposited on the silicon wafer

2. No validation tests are presented. It would be important to show, for example, tests on standard salts or blank spectra. Finally, it is absolutely necessary to carry out repeatability tests and to test the method under different conditions. I understand that the comparison with the literature on RH values can be seen as a comparison test, but in my view, this is not sufficient.

**Response:** To validate the effectiveness of this method, we employed it to determine the phase transition point during the hygroscopic growth of ammonium sulfate (AS) aerosols. Fig.S3(A) shows a synchronous PCMW2D correlation spectrum constructed from the RH-dependent IR spectra in the region of  $1502\sim1000~\rm cm^{-1}$  range during the humidification process of AS fine particles. From the synchronous correlation spectrum, it can be seen that (1097 cm-1, 80%) appears red, which is a positive correlation peak. It indicates that at 80% (RH), solid sulfate has transformed into liquid sulfate, and the deliquescence phase transition point is 80%. At the same time, ammonium ions also change most rapidly at this humidity condition (Fig.S3(B)). This suggests that AS of 300nm particles may have undergone deliquescence phase transition at  $80\% \pm 0.8\%$  (RH). Which is concordant with the results derived from

Fig.S3 Synchronous PCMW2D spectra in the (A)  $1502 \sim 1000$  cm-1 and (B)  $3550 \sim 2501$  cm-1 region during the humidification process of 300nm AS particles.

Fig.S4 Synchronous PCMW2D spectra in the  $3550 \sim 2501$  cm-1 during the humidification process of AS fine particles (A) 400nm AS particles; (B) 600nm AS particles.

Extended Aerosol Inorganics Model (E-AIM) (http://www.aim.env.uea.ac.uk/aim/aim.php.). Moreover, this result is consistent with the research results of Ahn et al. (2010)(Ahn et al., 2010), Cziczo et al. (1999)(Cziczo & Abbatt, 1999) and Li et al. (2017)(Li, F., C., P., & and Martin, 2017), who measured the deliquescence point of AS as  $79.9\% \pm 0.3\%$  and  $79\% \pm 2\%$  (in Table S1). The hydration characteristics of particles are governed primarily by their chemical

composition, and therefore the deliquescence point of AS is independent of particle size. So using this approach, we also analyzed the deliquescence phase transition of 400 nm and 600 nm AS particles during the deliquescence process and found that both of they were  $80\% \pm 0.8\%$  (RH) (Fig.S4). Therefore, this method is accurate in determining the phase transition point. We have included this explanation in the revised paper.

Table S1 Summary of deliquescence RH (DRH) and efflorescence RH (ERH) for AS from literature.

| DRH (%)        | ERH (%) | Ref                                                                          |
|----------------|----------------|------------------------------------------------------------------------------|
| $79.9 \pm 0.3$ | $37.9 \pm 0.5$ | Ahn et al. (2010) (Ahn et al., 2010)                                         |
| $79 \pm 1$     | 33± 2          | Cziczo et al. (1999)(Cziczo & Abbatt, 1999)                                  |
| 79.7           | 51.9 ~ 34.9    | Cai et al. (2017) (Cai et al., 2017)                                         |
| 80             | 35             | Schlenker et al. (2004)(Schlenker, Malinowski, Martin, Hung, & Rudich, 2004) |
| $79 \pm 2$     | /              | Li et al. (2017) (Li et al., 2017)                                           |
| /              | 38 ~ 40        | Yeung et al. (2009) (Yeung, Y., & and Chan, 2009)                            |

3. Regarding section 1.2, the presentation is a little too theoretical and not practical enough. For example, the size and overlap of the window used in PCMW2D are not specified, even though they directly affect the resolution and robustness of the results. Similarly, the description of spectrum pre-processing (baseline correction, noise correction, normalisation) is insufficient to guarantee reproducibility. Finally, no significance criteria are mentioned to distinguish real correlative peaks from noise. These details are necessary if we are to evaluate the robustness of the conclusions drawn from the 2D maps.

**Response:** Done. We have included a new paragraph, i.e., section 1.2, to present how we perform the size and overlap of the window used in PCMW2D. Significance criteria have been introduced to classify the signals. Specifically, signals below the 10% threshold are considered noise, while those above the 90% threshold are regarded as genuine correlation peaks. Please check the marked up file for details.

4. L146, it is stated that linear baseline corrections and smoothing are performed. Such preprocessing can have major implication on the signal. The paper should clearly

state which correction is applied and for which reason. In particular, I think that sensitivity tests should be done to show the impact on the conclusion.

Response: Done. ① We first corrected the baseline of the measured absorption spectra with the Opus 7.0 software. The linear baseline corrections were performed in the regions of 1000~1500cm-1 and 2500~3550cm-1 for all infrared spectra before calculation and analysis. ② Then we smoothed the noise of the measured absorption spectra. For the AS, the IR absorbance peaks (include the sulfate ion, ammonium ion and liquid water) are low-frequency, but the noise is random and high-frequency. A single scan could be smoothed to improve SNR. So the linear baseline corrections and smooth were performed in the regions of 1000~1500cm-1 and 2500~3550cm-1 for all infrared spectra before calculation and analysis. Due to the spectral resolution of 4 cm-1, the Savitzky-Golay filter with a smoothing width of 21 points was employed to effectively suppress noise while preserving the authentic line shapes of the characteristic infrared absorption bands. A comparative plot of the spectral data before and after processing is presented in Fig.S2, which shows the complete spectrum for raw and smoothed spectrum at 90% (RH).

Fig.S2 The complete spectrum for raw and smoothed spectrum at 90% (RH) of Ammonium Sulfate (AS) particles. Magenta rectangle indicates the spectral range that has been omitted from display in Fig. 1.

From Fig.1, the pre-processed infrared spectra clearly show the absorption peaks of sulfate, nitrate, and ammonium ions, and allows observation of their peak shifts with changes in relative humidity (RH).

I have some questions regarding the results and discussion. As these sections will likely need to be revised in light of my previous comments, the following comments should not be read as blocking, but rather as points that will likely need to be clarified later on:

1. Section 2.1: The authors restrict their analysis to the regions 1000–1500 and 2500–3550 cm-1, but do not show the complete spectrum. However, the evolution of other bands (e.g., lattice modes <900 cm-1, NH4+ bending ≈1680 cm-1) could provide additional clues about the liquid-solid transition and better support the proposed four-step sequence. I recommend presenting the complete FTIR spectra (before and after pre-treatment) in order to verify the absence of spurious peaks and identify any additional spectral signatures.

Response: Fig.S2 shows the complete spectrum for raw and smoothed spectrum at 90% (RH). The infrared absorption peaks in the 1500~2500 cm-1 range are primarily attributed to atmospheric water vapor and CO2. To better visualize the spectral variations of sulfate, ammonium, and water, we e we focused on the 1500–1000 cm-1 and 3500–2500 cm-1 regions, rather than showing the full spectrum. The bending vibration of NH4+ (1680 cm-1) falls within that of H2O (1400~1900 cm-1), with the absorption intensity of H2O's bending vibration being more pronounced. For the lattice modes < 900 cm-1, it could provide additional clues about the liquid-solid transition. While the infrared absorption signal is very weak and approaches the noise level. So we excluded the bending vibration of NH4+ and the lattice modes < 900 cm-1 from our analysis.

In future experiments, the following procedures will be implemented: the infrared spectrometer will be evacuated to eliminate interference from atmospheric CO2 and H2O, enabling accurate analysis of the NH4+ bending mode at 1680 cm-1; additionally,

the sample deposition mass will be increased to enhance signal intensity for effective examination of lattice modes below 900 cm-1.

Fig.S2 The complete spectrum for raw and smoothed spectrum at 90% (RH) of Ammonium Sulfate (AS) particles. Magenta rectangle indicates the spectral range that has been omitted from display in Fig. 1.

2. Typically, one would expect point 5 to be discussed in several sections, including 2.1: for example, does it have an impact on the estimated value of RH? Or when the authors state: 'Since asynchronous spectra reflect differing rates of spectral changes, the sequence of molecular bond transformations during efflorescence can be deduced as: (1084 cm-1) > (1097 cm-1) > (1463 cm-1) > (1417 cm-1).' (L306) How would preprocessing impact this point?

In this way, I would recommend to show the raw and post-processed results.

**Response:** Fig. R2 show the results from the 2D-IR correlation spectroscopic analysis of AS aerosol. In the synchronous 2D-IR correlation spectra, four strong autopeaks are observed at (1097, 1463) cm-1, (1417, 1463) cm-1, (1097, 1417) cm-1, and (1084, 1463) cm-1, indicating significant changes in these infrared absorption bands during the efflorescence phase transition. The cross-peak at (1097, 1463) cm-1 is positive. In the asynchronous 2D-IR correlation spectra, seven strong auto-peaks are present. Among them: (1417, 1463) cm-1, (1097, 1463) cm-1, (1084, 1417) cm-1, and (1097, 1084) cm-1 exhibit positive auto-peaks. While (1097, 1417) cm-1, (1084, 1463) cm-1, and (1084, 1097) cm-1 show negative auto-peaks. Since asynchronous

spectra reflect differing rates of spectral changes, the sequence of molecular bond transformations during efflorescence can be deduced as: The spectral variations at 1084 cm-1 appear earlier in the humidity-driven process than those at 1097 cm-1, which in turn precede the changes at 1463 cm-1, and finally, the variations at 1417 cm-1 occur last. Which is consistent with the results in Fig.3 (The post-processed results from the 2D-IR correlation spectroscopic analysis of AS aerosol in manuscript).

Fig. R2 The raw results from the 2D-IR correlation spectroscopic analysis of AS aerosol. (A) synchronous and (B) asynchronous 2D-IR correlation spectra of AS aerosol in the 1500-902 cm-1 region upon efflorescence. Red and blue color represent positive and negative correlations, respectively. RH ranges is  $90 \sim 20\%$  (RH), red and blue means a positive and negative correlation value, respectively.

A comparison between Fig. R2 (raw results) and Fig. 3 (post-processed results) reveals that the positions and colors of the synchronous and asynchronous cross-peaks are nearly identical, demonstrating that preprocessing has little effect on the results.

Fig.3 The post-processed results from the 2D-IR correlation spectroscopic analysis of AS aerosol. (A) synchronous and (B) asynchronous 2D-IR correlation spectra of AS aerosol in the 1500-902 cm-1 region upon efflorescence. Red and blue color represent positive and negative correlations, respectively. RH ranges is 90 ~ 20% (RH), red and blue means a positive and negative correlation value, respectively.

Fig. R3 Synchronous PCMW2D spectra in the 3550-2501 cm-1 region during the efflorescence process of AS aerosol. (A) The raw result (B) The post-processed result, which shows Fig. 2(C) from the manuscript.

From the synchronous correlation spectrum in Fig. R3(A), it can be seen that (3400 cm-1, 39%) appears blue. Which indicated that liquid water concentration diminished. This means the efflorescence phase transition point is  $39\% \pm 0.8\%$  (RH). It is consistent with that of the post-processed result. A comparison between Fig.R3(A) (raw results) and Fig.R3(B) (post-processed results) reveals that the positions and colors of t synchronous PCMW2D spectra are nearly identical, demonstrating that preprocessing has little effect on the results.

1. Discussion remains general and mostly hinge on visual observation. The interpretation would benefit from data-based interpretations, using statistical tests, spectrum analysis techniques etc. The values would then be thoroughly compared with literature, allowing for a proper validation of the method.

**Response:** In the revised version, we have presented proper comparisons and analysis with previous studies in discussion. Although AS efflorescence characteristics

observed in this study are similar to those in previous study (Cai et al., 2017; Cziczo & Abbatt, 1999; J. J. Nájera & Horn, 2009), The 2D-IR spectroscopic technique captured the complex phase transition processes and provided a more precise value for the efflorescence relative humidity (ERH) of ammonium sulfate (AS) aerosol. Please check discussion in the revised version for details.

2. I have probably missed one point, but when I look at the Figure 2, I would say that the maximum in correlation occurs at RH = 33 - 34 %. This is below the reported value of ( $39 \pm 0.8$ ) %. Can you clarify this point please? In the same way, I read maxima for the Fig 2B at RH = 32 and 35%. Section 2.2 needs a similar revision.

**Response:** This is caused by a misrepresentation of the x-axis values. We have redrawn Fig.2 in manuscript following our reanalysis of the PCMW2D spectra of AS aerosol. The efflorescence phase transition point is  $39\% \pm 0.8\%$  (RH). Please check the marked up file for details.

Finally, I would recommend using more cautious language and not overinterpreting the results. If the method is sufficiently reliable, it should speak for itself, and the overly superlative language disturbed me much more than it convinced me of the method's merits. Here are some examples, but again, it is true for most of the paper.

• "This study could provide critical insight about redefining atmospheric heterogeneous chemistry." (L87)

**Response:** Done. Please check the marked up file for details.

• Lines 293-298: "Conversely, three negative cross-peaks [...] causing these ions to move closer together." The authors directly interpret the negative cross-peaks as evidence that ammonium and sulphate ions move closer together during water loss. However, in my opinion, a negative cross-peak only indicates that the spectral intensities at the two wavelengths vary in opposite directions with RH. The translation into structural terms ("ions move closer together") is a plausible hypothesis, but it should be formulated more cautiously and supported by other evidence (e.g., band shifts (section 2.1) or comparison with previous work).

**Response:** The observed red-shift of the SO42- stretching band can be explained by the stronger hydrogen bonding between SO42- and NH4+ compared to that between SO42- and H2O. As H2O molecules in the hydration shell are replaced by NH4+ ions during crystallization, the effective mass of the vibrating oxygen atoms increases, leading to the observed frequency decrease. In the revised version, we have added this explanation based on your suggestion. Please check the marked up file for details.

• Similarly, the presentation of the sequence of AS efflorescence (1084 > 1097 > 1463 > 1417 cm-1) might be too strong. A more appropriate phrasing would be that the spectral variations at 1084 cm-1 appear earlier in the humidity-driven process than those at 1097 cm-1, and so on. The physical interpretation of these sequences (e.g. as bond dissociation or ion rearrangements) should remain a hypothesis rather than a demonstrated fact. What the asynchronous 2D-COS actually shows is that spectral variations at 1084 cm-1 occur earlier in the RH-driven process than those at 1097 cm-1, and so on. I recommend reformulating in that sense, to avoid over-interpretation.

**Response:** Done. Please check the marked up file for details.

• Following the latter point, the concluding paragraph of section 2.3 should be revised. The mechanistic details proposed (hydrogen bond dissociation, ions approaching) go beyond what FTIR/2D-COS can directly demonstrate, and should be framed as hypotheses rather than conclusions. The final statement on NH4+ surface enrichment is particularly problematic: this cannot be inferred from the present data and appears to be borrowed from Tian et al. (2011). I recommend the authors either remove this claim or clearly state it as literature context, not as a conclusion of their own work.

**Response:** Done. Please check the marked up file for details.

• Section 2.4 concludes that the microscopic kinetic evolution of AS efflorescence is elucidated 'at the molecular level.' This phrasing is too strong. 2D-COS analysis does reveal sequential spectral changes and can suggest a two-step efflorescence pathway, but it does not directly provide molecular-level kinetics. Such interpretations require careful assumptions and should be stated more cautiously, e.g.

as 'the data are consistent with a multi-step efflorescence mechanism' rather than as a direct elucidation of molecular dynamics.

**Response:** Done. Please check the marked up file for details.

More minor mistakes are present along the manuscript such as English wording, capital letters or citation format. Although this is not in itself problematic, these small issues accumulate and could distract readers. Careful proofreading would help improve the overall readability and presentation.

**Response:** Done. Please check the marked up file for details.

In conclusion, I think this manuscript illustrates the difficulty of writing methodfocused papers, which require a careful balance between presenting the technique itself
and showing how it is validated. In its present form, the introduction and framing
currently do not meet the standards expected for AMT. This, in my view, is less a
reflection of the first author's effort than of the need for closer guidance from the
supervisors in positioning the work. With stronger supervision and restructuring, I
believe this study could evolve into a much more solid contribution, and I encourage
the authors to pursue this line of research.

**References:**

- Ahn, K.-H., Kim, S.-M., Jung, H.-J., Lee, M.-J., Eom, H.-J., Maskey, S., & Ro, C.-U. (2010). Combined Use of Optical and Electron Microscopic Techniques for the Measurement of Hygroscopic Property, Chemical Composition, and Morphology of Individual Aerosol Particles. *Analytical Chemistry*, 82(19), 7999-8009. doi:10.1021/ac101432y
- Cziczo, D. J., & Abbatt, J. P. D. (1999). Deliquescence, efflorescence, and supercooling of ammonium sulfate aerosols at low temperature: Implications for cirrus cloud formation and aerosol phase in the atmosphere. *JOURNAL OF GEOPHYSICAL RESEARCH-ATMOSPHERES*, 104(D11), 13781-13790. doi:10.1029/1999JD900112
- Li, Y. J., F., L. P., C., B., P., B. A., & and Martin, S. T. (2017). Rebounding hygroscopic inorganic aerosol particles: Liquids, gels, and hydrates. *Aerosol Science and Technology*, 51(3), 388-396. doi:10.1080/02786826.2016.1263384
- Schlenker, J. C., Malinowski, A., Martin, S. T., Hung, H.-M., & Rudich, Y. (2004). Crystals Formed at 293 K by Aqueous Sulfate–Nitrate–Ammonium–Proton Aerosol Particles. *The Journal of Physical Chemistry A*, 108(43), 9375-9383.

doi:10.1021/jp047836z

Yeung, M. C., Y., L. A. K., & and Chan, C. K. (2009). Phase Transition and Hygroscopic Properties of Internally Mixed Ammonium Sulfate and Adipic Acid (AS-AA) Particles by Optical Microscopic Imaging and Raman Spectroscopy. *Aerosol Science and Technology*, 43(5), 387-399. doi:10.1080/02786820802672904

---

## Author Response (AR2)

**Response to Referee #2:**

Thanks very much for your comments, suggestions and recommendation with respect to improve this paper. The response to all your comments is listed below.

In generally, I would like to accept this article to be published in AMT after one minor revision. Here I also have several questions on this paper improvement. Seemly, this paper is too technique. I would like to suggest more extend and embrace more references on this paper. As I knew, there are more studies working on phase transitions of aerosol particles. The authors should emphasize more literature and extend your study for the potential uses in the campaign.

For example,

Sun, et al., Key Role of Nitrate in Phase Transitions of Urban Particles: Implications of Important Reactive Surfaces for Secondary Aerosol Formation, J. Geophy. Res., 123, 1234-1243, 10.1002/2017JD027264, 2018.

Wise, et al., (2008). Water uptake by NaCl particles prior to deliquescence and the phase rule. Aerosol Science and Technology, 42, 281–294. https://doi.org/10.1080/02786820802047115

Li, et al., Microscopic Evidence for Phase Separation of Organic Species and Inorganic Salts in Fine Ambient Aerosol Particles, Environ. Sci. Techn., 55, 2234-2242, 10.1021/acs.est.0c02333, 2021.

There are more studies using ETEM, ESEM, and optical microscopicy.

**Response:** We have followed your suggestion and integrated some key references on aerosol phase transitions, which has strengthened the scientific context of our work. Please check the marked up file for details.